# Scattering of e^±^ by C_2_H_6_ Molecule over a Wide Range of Energy: A Theoretical Investigation

**DOI:** 10.3390/molecules28031255

**Published:** 2023-01-27

**Authors:** N. M. B. Sathee, M. Mousumi Khatun, Anita Rani, M. Masum Billah, M. Nure Alam Abdullah, Mahmudul H. Khandker, Hiroshi Watabe, A. K. Fazlul Haque, M. Alfaz Uddin

**Affiliations:** 1Atomic and Molecular Physics Research Laboratory, Department of Physics, University of Rajshahi, Rajshahi 6205, Bangladesh; 2Institute of Fuel Research and Development (IFRD), Bangladesh Council of Scientific and Industrial Research (BCSIR), Dhaka 1205, Bangladesh; 3Department of Physics, Jagannath University, Dhaka 1100, Bangladesh; 4Division of Radiation Protection and Safety Control, Cyclotron and Radioisotope Center, Tohoku University, Sendai 980-8578, Japan

**Keywords:** electron and positron scattering, molecular scattering, C_2_H_6_, independent atom model, screening correction

## Abstract

The present work reports the theoretical investigation of the scattering of electrons and positrons by the ethane (C${}_{2}$H${}_{6}$) molecule over the energy range 1 eV–1 MeV. The investigation was carried out by taking into account the screening correction arising from a semiclassical analysis of the atomic geometrical overlapping of the scattering observables calculated in the independent atom approximation. The study is presented through the calculations of a broad spectrum of observable quantities, namely differential, integrated elastic, momentum transfer, viscosity, inelastic, grand total, and total ionization cross-sections and the Sherman functions. A comparative study was carried out between scattering observables for electron impact with those for positron impact to exhibit the similarity and dissimilarity arising out of the difference of the collisions of impinging projectiles with the target. Partial-wave decomposition of the scattering states within the Dirac relativistic framework employing a free-atom complex optical model potential was used to calculate the corresponding observable quantities of the constituent atoms. The results, calculated using our recipe, were compared with the experimental and theoretical works available in the literature. The Sherman function for a $e^{\pm }$–$C_{2}H_{6}$ scattering system is presented for the first time in the literature. The addition of the screening correction to the independent atom approximation method was found to substantially reduce the scattering cross-sections, particularly at forward angles for lower incident energies.

## 1. Introduction

The scattering of electron ($e^{-}$) and positron ($e^{+}$) by molecular targets has drawn considerable interest in the recent past as it provides the basic understanding of their transport phenomena and interaction processes with the atoms and nuclei of the target, as well as important insights into the collision dynamics and molecular structure. The knowledge of $e^{\pm }$–molecule scattering is requisite in various fields of science and technology such as space science, plasma physics, atmospheric physics, astrophysics and chemical physics [1,2,3,4,5]. The $e^{\pm }$–molecule scattering cross-section data play a key role in plasma processes, gaseous dielectrics, discharge switches, gas lasers, space science, radiation research, etc. In magnetic fusion energy research, the atomic and molecular scattering data are used (i) to make the vessel walls insulated from the hot plasma core in the cooler edge plasma and (ii) to remove the impurity waste materials in magnetic divertors [6]. Accurate cross-sections for $e^{-}$–molecule scattering at low and intermediate energies obtained from theoretical models help to understand the interaction process. Moreover, the cross-section for high-energy electrons helps to study the contribution of binding electrons to the scattering potential [7].

The positron, since its discovery [8,9,10], is being used as the probe in many experiments of fundamental and applied sciences, which led to the discovery of positronium and the positronium ion. This also led to the first production tapping and spectroscopy of antihydrogen. Low-energy positron beams are used to study the structure of matter on the microscopic scale. The positron annihilation technique is used to yield the information of several properties related to the internal arrangement of material defects [11]. Since positrons are very sensitive to crystal defects, the positron annihilation method is also used to investigate the defect properties of solids. The process of positronium formation and its annihilation is of special interest in astrophysical problems. This leads to the production of solar flares with the emission of γ-rays from the Sun [12]. Moreover, a comprehensive dataset of positron impact molecular scattering cross-sections is required in astrophysical research, radiation-based technologies, and energy deposition models [2]. The exchange potential is important for electron scattering, which is absent for the case of positron scattering. However, the polarization effect is substantial for positron scattering. This is due to the cancellation effects of static and polarization potentials because the former is a repulsive and the latter is an attractive potential.

In molecular physics, hydrocarbon molecules such as ${\mathrm{CH}}_{4}$, $C_{2}H_{2}$, $C_{2}H_{4}$, and $C_{2}H_{6}$, etc., play an important role as they are prototypes of polyatomic molecules. Ethane ($C_{2}H_{6}$) is an important tracer in the atmosphere of Saturn and Jupiter, as it absorbs the infrared radiation in the atmosphere of these planets [13]. Reliable elastic scattering cross-section data for $C_{2}H_{6}$ are, therefore, essential for planetary modeling [5]. Accurate cross-sections for hydrocarbons are also required in fusion plasma devices as these are produced during the reaction between graphite and deuterium fuel [14]. Hydrocarbon molecules are also known to be dominant materials in plasma processing, where these are used in the edge plasma magnetically confined with hydrogen plasma at high temperature. Being a member of homologous series of compounds, $C_{2}H_{6}$ has an important role in green chemistry. In chemical deposition industries, hydrocarbons are used to produce high-quality solid materials in which carbon, produced in a low-temperature discharge, hardens the substrates.

The scattering of $e^{-}$ and $e^{+}$ by the $C_{2}H_{6}$ molecule has been studied by several experimental [15,16,17,18,19,20,21,22,23,24,25,26,27,28,29,29] and theoretical [23,27,28,30,31,32,33,34] groups. Fink et al. [16] measured the elastic DCSs for the electron scattering by $C_{2}H_{6}$, $C_{2}H_{4}$, and $C_{2}H_{2}$ in the energy range $E_{i}=100$–1000 eV. The data of Fink et al. [16] are normalized with our calculations at angle $100^{\circ }$. Tanaka et al. [22] performed an experiment to measure the absolute DCSs for electron energies of 2–100 eV using a relative flow technique of a crossed-beam type apparatus. In their work, Rawat et al. [23] reported an experimental measurement of absolute cross-sections for $e^{-}$–$C_{2}H_{6}$ scattering at impact energies between 40 and 500 eV using a crossed electron beam–molecular beam geometry technique. They also made theoretical calculations on the differential cross-section (DCS), integrated elastic cross-section (IECS), momentum transfer cross-section (MTCS), total cross-section (TCS), and total absorption cross-section for electron energies in the range 1–500 eV using a complex optical model potential (OMP). Mapstone and Newell [24] measured the elastic DCS for $C_{2}H_{6}$ at incident energies $E_{i}=3.2$–15.4 eV using a hemispherical electron spectrometer. The elastic DCS for the electron scattering from CO, ${\mathrm{CO}}_{2}$, ${\mathrm{CH}}_{4}$, $C_{2}H_{4}$, and $C_{2}H_{6}$ in the energy range 300–1300 eV was measured by Maji et al. [27] using the crossed-beam technique. They [27] also calculated the above cross-sections in terms of the independent atom model (IAM). The absolute elastic and inelastic DCSs for the scattering of $e^{-}$ from ${\mathrm{CH}}_{4}$ and $C_{2}H_{6}$ were measured by Curry et al. [20] for projectile energies ranging from 7.5 to 20 eV using a double-hemispherical electron spectrometer. The experimental measurement of TCSs for the scattering of electrons by the $C_{2}H_{6}$ molecule are abundant in the literature [19,21,25,26,29] up to 500 eV incident energy. The total ionization cross-section (TICS) for $C_{2}H_{6}$ was measured by [35,36,37,38,39] for electron energies up to 12 keV. Shishikura et al. [40] derived the MTCS up to 100 eV from their measurement of drift velocity and the longitudinal diffusion coefficient of electrons. Duncan et al. [15] evaluated the MTCS from their D/μ (the ratio of the lateral diffusion coefficient to electron mobility) measurement for the scattering of electrons by some hydrocarbon molecules including $C_{2}H_{6}$ in the energy range 0.01–1.0 eV.

Bettega et al. [33] theoretically investigated the DCS and IECS for electron scattering from $B_{2}H_{6}$, $C_{2}H_{6}$, ${\mathrm{Si}}_{2}H_{6}$, and ${\mathrm{Ge}}_{2}H_{6}$ from 5 to 30 eV using an ab initio Schwinger multichannel method (SMC) with pseudopotentials. In another work, Bettega et al. [41] calculated the elastic DCS and IECS for low-energy electron scattering up to 12 eV employing the same SMC including the polarization effect with the pseudopotential. The TCS and inelastic cross-sections (INCS) for $e^{-}$–$C_{2}H_{6}$ scattering were theoretically studied by Joshipura and Vinodkumar [42] at collision energies $E_{i}$ = 50–5000 eV using optical model potential (OPM) with the additivity rule (AR), wherein the molecular cross-section was an incoherent sum of the cross-sections of the constituent atoms. Using the AR, Jin-Feng et al. [43] calculated the TCS of electrons for hydrocarbon molecules including $C_{2}H_{6}$ at impact energies 10–2000 eV. Vinodkumar et al. [44] reported the theoretical calculation of the TCS (50–2000 eV) and TICS (threshold to 2000 eV) for $C_{2}H_{6}$ in terms of the spherical complex optical potential (SCOP) and complex scattering potential-ionization contribution (CSP-ic) methods. Using the binary-encounter Bethe (BEB) model combining the binary-encounter and Bethe theories, Hwang et al. [45] theoretically investigated the electron TICS for $C_{2}H_{6}$ up to 10,000 eV. Sun et al. [32] studied the IECS for low-energy $e^{-}$ scattering by $C_{2}H_{6}$ in terms of ab initio theory employing the complex Kohn method. They found a Ramsauer–Townsend (RT) minimum at a 0.18 eV electron energy, which bears a non-zero quadrupole moment. M. Hayashi [46] reported the MTCSs for $C_{2}H_{6}$ up to 1000 eV of incident energy.

The experimental measurement of $e^{+}$–$C_{2}H_{6}$ scattering is, however, limited in the literature to date. Chiari et al. [28] reported the experimental measurement of the TCS and the theoretical investigation of the elastic DCS and IECS at positron energies 1–10 eV. Sueoka and Mori [21] measured the TCS for positron scattering by $C_{2}H_{6}$ for energies ranging from 0.7 to 400 eV. The TCS of some hydrocarbons with $C_{2}H_{6}$ were measured by Floeder et al. [19] for positron energies between 5 and 400 eV. Raizada and Baluja [47] theoretically investigated the TCS for the scattering of $e^{+}$ from $C_{2}H_{6}$ using the AR in terms of OMP up to 5000 impact energy. The TCS for $e^{+}$–$C_{2}H_{6}$ scattering in the energy range 100–1000 eV was also calculated by Raj and Tomar [48] using the independent atom model (IAM). Recently, Singh and Antony [49], in 2018, used the modified SCOP and CSP-ic methods, respectively, to calculate the positron TCS and TICS for $C_{2}H_{6}$ at collision energies from the positronium threshold to 5000 eV. Occhigrossi and Gianturco [50] theoretically calculated the low-energy positron IECS at impact energies up to 6 eV. To the best of our knowledge, data, both experimental and theoretical, on viscosity cross-section (VCS) for this collision system is not available in the literature.

Although there is a good number of studies found in the literature for the scattering of electrons by $C_{2}H_{6}$, none of them are on the Sherman function S(θ). Moreover, the works on DCS are limited to 1300 eV incident energy. For positron scattering, on the other hand, there is only one work in the literature [28], to the best of our knowledge, for the DCS, which was limited to 10 eV. The calculation of scattering observables for this molecule therefore has importance.

As mentioned earlier, reliable data on $e^{\pm }$–molecule scattering is important in many areas of science and technology. The extraction of the experimental data for $e^{\pm }$–molecule collisions involves complicated experimental procedures. As such, sometimes, the data suffer from errors in the magnitudes that are beyond the experimental uncertainties [26]. Hence, despite the availability of experimental data, models, based on sound theory, can be useful in deriving consistent cross-sections for different scattering observables, which can overcome the ambiguities persisting in experimental data and also help to furnish future experiments. Moreover, the literature lacks a single theoretical model, capable of predicting a wide spectrum of scattering observables over a wide incident energy range, particularly for the $e^{\pm }$–C${}_{2}$H${}_{6}$ scattering system. The Dirac partial wave theory (DPWT) in the framework of the optical model (OM) potential was used in the present work, which inherently includes the spin–orbit term, which is essential in explaining the spin polarization [51,52,53]. The complex OM potential consists of the static, the exchange (absent for positrons), the polarization correlation, and the imaginary terms. The static potential has a major contribution on the scattering observables [53]. The polarization correlation potential, which has an effect at low incident energies and small scattering angles, is taken into account in the OM potential as the electric field of the incident particle affects the charge distribution of the target atom, which causes the polarization of the charge cloud. The occurrence of rearrangement collision between the incident electron and atomic orbital electrons is addressed by the exchange potential. The imaginary term was added in the OM potential to account for the loss of flux from the elastic to the inelastic channels.

The collision dynamics of $e^{\pm }-$molecule is more complicated than that of an atom because of the multi-center nature and non-spherical charge distribution of the molecule. There are also some additional degrees of freedom in a molecule such as rotation and vibration, for which there arises more complexity in deriving molecular cross-sections. Therefore, the calculation of $e^{\pm }$–molecule scattering cross-sections is challenging and more complicated than that of $e^{\pm }$–atom scattering. It is important to choose the correct form of potential to calculate different scattering observables such as DCS, TCS, TICS, MTCS, VCS, IECS, INCS, and S(θ).

This paper is presented as follows. The mathematical details are discussed in Section 2. The results of our proposed model and the comparison of our results with the available data are presented in Section 3. In Section 4, we draw the conclusions about our results.

## 2. Outline of the Theory

### 2.1. Interaction Potential

In the OPM, the interaction potential between the incident $e^{\pm }$ and the constituent atoms of the molecular target can be expressed as [54]:(1)$$V_{OPM}\left(r\right)=V_{st}\left(r\right)+V_{ex}\left(r\right)+V_{cp}\left(r\right)-iW_{abs}\left(r\right),$$
with $V_{ex}=0$ for the positron. In Equation (Equation 1), the first, second, third, and fourth terms are, respectively, the static potential, the exchange potential, the correlation polarization potential, and the imaginary potential.

The static potential $V_{st}\left(r\right)$ due to the charge distribution of the constituent atoms is given [55] by
(2)$$V_{st}\left(r\right)=\pm e\phi \left(r\right)=\pm e\left[\phi_{n}\left(r\right)+\phi_{e}\left(r\right)\right],$$
where *e* is the charge of the projectile (the + sign is used for positrons and the − sign for electrons) and $\phi_{n}\left(r\right)$ and $\phi_{e}\left(r\right)$ are, respectively, the contributions from the nucleus and electron cloud of the target. The Dirac–Fock electron density, generated within the framework of the multi-configuration technique by the Desclaux code [56], was used in the present study.

The exchange potential, used in this study, is the semi-classical local exchange potential of Furness and McCarthy [57], derived from the non-local exchange interaction using the Wentzel-Kramers-Brillouin (WKB) approximation for the wavefunctions and is given by
(3)$$V_{ex}\left(r\right)=\frac{1}{2}\left[E_{i}-V_{st}\left(r\right)\right]-\frac{1}{2}{\left\{{\left[E_{i}-V_{st}\left(r\right)\right]}^{2}+4\pi a_{0}e^{4}\varrho_{e}\left(r\right)\right\}}^{1/2},$$
where $E_{i}$ and $a_{0}$ are, respectively, the kinetic energy of the incident electron and the Bohr radius of the target atom.

The combination of a short-range local density approximation (LDA) correlation potential $V_{co}\left(r\right)$ and a long-range Buckingham potential $V_{p}$ is used as the correlation-polarization potential, which is expressed as [54]
(4)$$V_{cp}^{\pm }\left(r\right)\equiv \left\{\begin{array}{cc} \max\{V_{co}^{\pm }\left(r\right),V_{p}\left(r\right)\} & \;\mathrm{if}\;\;r<r_{cp}\\ V_{p}\left(r\right) & \;\mathrm{if}\;\;r\geq r_{cp}. \end{array}\right.$$
Here, $r_{cp}$ is the outer radius at which $V_{co}^{\pm }\left(r\right)$ crosses first with $V_{p}$. In Equation (Equation 4), $V_{cp}^{-}\left(r\right)$ is used for electrons [58] and $V_{cp}^{+}\left(r\right)$ for positrons [59].

For the imaginary potential, the following semi-relativistic form proposed by Salvat et al. [55] is used, which is given by
(5)$$W_{abs}\left(r\right)=A_{abs}\frac{\hbar }{2}\left[v_{L}^{nr}\varrho_{e}\left(r\right)\sigma_{bc}\left(E_{L},\varrho_{e},\Delta_{e}\right)\right]{\left[\frac{2{\left(E_{L}+m_{e}c^{2}\right)}^{2}}{m_{e}c^{2}\left(E_{L}+m_{e}c^{2}\right)}\right]}^{\frac{1}{2}}.$$

Here, $m_{e}$ and *c* are, respectively, the mass of the projectile and the velocity of light in free space. $A_{abs}$ is a projectile–target dependent parameter, which has the value $A_{abs}=2.0$ for both electrons and positrons.

### 2.2. Partial Wave Analysis

In the IAM, the direct and spin flip amplitudes for the scattering of $e^{\pm }$ from the $C_{2}H_{6}$ molecule for a particular orientation can be written [60] as
(6)$$F\left(\theta \right)=\sum_{i}\exp\left(iq.r_{i}\right)f_{i}\left(\theta \right),$$
and
(7)$$G\left(\theta \right)=\sum_{i}\exp\left(iq.r_{i}\right)g_{i}\left(\theta \right),$$
with $f_{i}\left(\theta \right)$ and $g_{i}\left(\theta \right)$ being, respectively, the direct and spin flip amplitudes for the constituent atoms of the molecule. ℏq and $r_{i}$, in Equations (Equation 6) and (Equation 7), respectively, represent the momentum transferred during the collision and the nuclear position vector of the *i*-th atom relative to an arbitrary origin. There is scope for 25,000 partial waves. Angular momentum is increased up to a certain maximum for which the absolute value of the phase-shift is less than $10^{-9}$. At this point, partial wave expressions for $f_{i}\left(\theta \right)$ and $g_{i}\left(\theta \right)$ converge to the required accuracy (usually more than six decimal places) for all angles [55].

Since the molecule rotates, the corresponding differential cross-section is obtained by averaging over all the orientations of the molecular axis:(8)$$\frac{d\sigma }{d\Omega }={\langle |F\left(\theta \right)|}^{2}+{\left|G\left(\theta \right)\right|}^{2}\rangle$$Use of Equations (Equation 6) and (Equation 7) in Equation (Equation 8) gives
(9)$$\begin{array}{c} \frac{d\sigma }{d\Omega }=\sum_{i}\left[\right|f_{i}{\left(\theta \right)|}^{2}+|g_{i}\left(\theta \right){|}^{2}]+\sum_{i\neq j}\frac{\sin(qr_{ij})}{qr_{ij}}\left[f_{i}\left(\theta \right)f_{j}^{*}\left(\theta \right)+g_{i}\left(\theta \right)g_{j}^{*}\left(\theta \right)\right], \end{array}$$
where q=2ksin(θ/2), $r_{ij}$ is the distance between the *i*-th and *j*-th atoms of the target molecule, $\sin\left(qr_{ij}\right)/qr_{ij}=1$ when $qr_{ij}=0$, and the term $\sum_{i\neq j}$ represents the contribution of the interference effect to the molecular DCS.

In terms of the DCS, the IECS, MTCS, and VCS for the projectile–molecule collision can, respectively, be expressed as
(10)$$\sigma_{el}=\int \frac{d\sigma }{d\Omega }d\Omega =2\pi \int_{0}^{\pi }\frac{d\sigma }{d\Omega }\sin\theta d\theta ,$$
(11)$$\sigma_{m}=2\pi \int_{0}^{\pi }\left(1-\cos\theta \right)\frac{d\sigma }{d\Omega }\sin\theta d\theta ,$$
and
(12)$$\sigma_{v}=3\pi \int_{0}^{\pi }\left[1-{\left(\cos\theta \right)}^{2}\right]\frac{d\sigma }{d\Omega }\sin\theta d\theta .$$The TCS for the *i*-th atom is the sum of the corresponding integrated elastic and inelastic cross-sections and is given as [61,62]
(13)(σtot)i=(σel)i+(σinel)i=4πkImfi(0),
where Imfi(0) is the imaginary part of $f_{i}\left(\theta \right)$ at θ=0.

To account for the mutual overlapping of nearby atoms in a molecule, Blanco and Garcia [63] proposed a screening correction. According to a semi-classical analysis [63], the screening correction coefficients $s_{i}$ ($0\leq s_{i}\leq 1$) for the *i*-th atom of a molecule can be given as a sum of $\varepsilon_{i}^{\left(m\right)}$ terms, each of them arising from m-atoms overlapping:(14)$$s_{i}=1-\frac{\varepsilon_{i}^{\left(2\right)}}{2!}+\frac{\varepsilon_{i}^{\left(3\right)}}{3!}-\frac{\varepsilon_{i}^{\left(4\right)}}{4!}+......\pm \frac{\varepsilon_{i}^{\left(N\right)}}{N!}$$
where
(15)$$\varepsilon_{i}^{\left(m\right)}=\frac{N-m+1}{N-1}\sum_{i\neq j}\frac{\sigma_{j}\varepsilon_{j}^{\left(m-1\right)}}{\alpha_{ij}}\;\left(m=2,.....,N\right).\;$$Here, *N* is the number of atoms in the target molecule, the *j* index in sums $\sum_{j(\neq 1)}$ runs over all the N atoms except the *i*-th one, $\alpha_{ij}=max\left(4\pi r_{ij}^{2},\sigma_{i},\sigma_{j}\right)$, $\sigma_{i}$ is the atomic cross-sections for the *i*-th atom of the molecule, and $r_{ij}$ is the distance between the centers of atoms *i* and *j*. The coefficients $s_{i}$ refrain from counting each electron interaction with a pair of overlapped atoms twice. Another factor $\nu_{ij}$ is defined, to quantify the screening corrections to the interference contributions, as $\nu_{ij}=r_{ij}^{2}/\left(r_{ij}^{2}+\rho_{ij}^{2}\right)$ with $\rho_{ij}=max\left(\sqrt{\sigma_{i}/\pi },\sqrt{\sigma_{j}/\pi },1/k\right)$ being a length-dimensional parameter [64]. Since $\left(\sqrt{\sigma /\pi }\right)$ corresponds to the radius of a circle of area σ, the condition $r_{ij}=max\left(\sqrt{\sigma_{i}/\pi },\sqrt{\sigma_{j}/\pi }\right)$ represents a situation of geometrical overlap between two disks for which the center of the smallest one approaches the border of the other. The screening-corrected version of Equation (Equation 9) can now be written as
(16)$$\begin{array}{cc} {\left(\frac{d\sigma }{d\Omega }\right)}^{s}= & \sum_{i}s_{i}^{2}\left[\right|f_{i}{\left(\theta \right)|}^{2}+|g_{i}\left(\theta \right){|}^{2}]\\ & +\sum_{i\neq j}\nu_{ij}s_{i}s_{j}\frac{\sin(qr_{ij})}{qr_{ij}}\left[f_{i}\left(\theta \right)f_{j}^{*}\left(\theta \right)+g_{i}\left(\theta \right)g_{j}^{*}\left(\theta \right)\right] \end{array}$$

The first summation in Equation (Equation 16) accounts for each atomic contribution, reduced by the $s_{i}$ factors and the second one for the reduced interference contributions. The screening-corrected integrated elastic $\sigma_{el}^{s}$, momentum transfer $\sigma_{m}^{s}$, and viscosity $\sigma_{v}^{s}$ cross-sections are obtained from Equations (Equation 10)–(Equation 12) replacing $\frac{d\sigma }{d\Omega }$ with ${\left(\frac{d\sigma }{d\Omega }\right)}^{s}$ from Equation (Equation 16). The screening-corrected total $\sigma_{tot}^{s}$ cross-section is given by
(17)$$\sigma_{tot}^{s}=\sigma_{el}^{s}+\sigma_{inel}^{s}=\sum_{i}\;s_{i}\left(\sigma_{el}+\sigma_{inel}\right)=\sum_{i}\;s_{i}\sigma_{tot}.$$

The spin polarization or the Sherman function of the randomly oriented molecule is given by
(18)$$\begin{array}{c} S\left(\theta \right)=i\frac{\langle F\left(\theta \right)G^{*}\left(\theta \right)-F^{*}\left(\theta \right)G\left(\theta \right)\rangle }{\langle |F\left(\theta \right){|}^{2}+|G\left(\theta \right){|}^{2}\rangle }. \end{array}$$

The scattering amplitude from the *i*-th atom is calculated using the following form of the effective polarizability:(19)$$\alpha_{d,\mathrm{eff}}\left(i\right)=\alpha_{d}^{mol}\alpha_{d}\left(i\right){\left[\sum_{j}\alpha_{d}\left(j\right)\right]}^{-1}.$$Here, the summation extends over all the constituent atoms in the target $C_{2}H_{6}$. We used the values $\alpha_{d}=1.76\;{\overset{˚}{A}}^{3}$ for carbon and $\alpha_{d}=0.666793\;{\overset{˚}{A}}^{3}$ for hydrogen as given in [65] for the atomic dipole polarizabilities. In the IAM approximation, the long-range polarization potential $V_{p}\left(r\right)$ for an atom is calculated using the effective atomic polarizability defined in Equation (Equation 19).

The total inelastic cross-section $\sigma_{inel}$ can be partitioned into two main contributions:(20)$$\sigma_{inel}\left(E_{i}\right)=\sum \sigma_{exc}\left(E_{i}\right)+\sigma_{tion}\left(E_{i}\right),$$
where the first term is the sum over the total excitation cross-sections and the second term is the total ionization cross-section. Now, the total ionization cross-section is obtained by adding the direct ionization cross-section with the positronium formation cross-section being its respective energies, such that $\sigma_{tion}\left(E_{i}\right)=\sigma_{ion}\left(E_{i}\right)+\sigma_{ps}\left(E_{i}\right)$. Positronium formation, the short-lived exotic hydrogen like an atom, takes place at low and intermediate energies as a result of the interaction of the positron with the bound electrons of the target. Our method lacks the option for the treatment of the positronium formation and annihilation. The first term in Equation (Equation 20) becomes less and less important than the second at energies well above the ionization threshold. Hence, we can write:(21)$$\sigma_{inel}\geq \sigma_{tion}.$$The TICS $\sigma_{ion}$ can now be calculated from the following ratio [66]:(22)$$R\left(E_{i}\right)=\frac{\sigma_{ion}\left(E_{i}\right)}{\sigma_{inel}\left(E_{i}\right)}$$
with $0\leq R(E_{i})\leq 1$. For projectile energies greater than the ionization potential $\left(E_{i}>I\right)$, the ratio $R(E_{i})$ can be fit to the equation:(23)$$R\left(E_{i}\right)=1-C_{1}\left[\frac{C_{2}}{U+A}+\frac{\ln U}{U}\right]$$
where $U=E_{i}/I$ is the reduced energy. The adjustable parameters $C_{1}$, $C_{2}$ and *A* are determined using the following conditions.
(24)$$R\left(E_{i}\right)=\left\{\begin{array}{cc} 0\;\; & \mathrm{for}\;\;E_{i}\leq I\\ R_{p} & \mathrm{for}\;\;E_{i}=E_{p}\\ R_{F}\;\; & \mathrm{for}\;\;E_{i}\geq E_{F}>E_{P}. \end{array}\right.$$The first condition of Equation (Equation 24) implies that no ionization takes place below the ionization threshold energy of the molecule. Here, $E_{p}$ is the impact energy at which absorption obtains its maximum and $R_{p}$ represents *R* at $E_{i}=E_{p}$. At incident energies $E_{i}\geq E_{F}$, beyond the peak position $E_{p}$, the value of *R* increases to $R_{F}$ (very close to 1). The optimum values of the parameters $C_{1}$, $C_{2}$, and *A* are obtained from the solutions of Equation (Equation 24) using a FORTRAN program, which are $R_{P}=0.83$, $E_{P}$= 80 eV, $R_{F}=0.98$, $E_{F}$= 900 eV, $C_{1}=-1.583$, $C_{2}=-5.917$, and A=8.369 for both projectiles. More details of the theory of scattering by complex potentials is available in [55,61,62].

## 3. Results and Discussion

In the present work, the Elastic Scattering of Electrons and Positrons by Atoms (ELSEPA) code [55] was used to calculate various scattering observables for the $e^{\pm }$–$C_{2}H_{6}$ scattering system, over the energy range 1eV–1MeV using both the IAM and IAMS. The target properties and the coordinate geometry, used in the calculation of observable quantities, are presented in Table 1 and Table 2. Most of our calculations were performed in Hartree atomic units in which $\hbar =m_{e}=e=1$. If partial wave analysis is feasible, the calculated DCSs, integrated cross-sections, and Sherman function are usually accurate to within about 0.01%. This error estimate refers only to the accuracy of the numerical calculation and is based on the assumption that the adopted central potential represents the true interaction [55]. In the following subsections, the results of the calculated DCS, S(θ), TCS, TICS, IECS, INCS, MTCS, and VCS are presented and discussed for both electron and positron scattering.

### 3.1. Differential Cross-Section and Sherman Function S(θ)

The calculated results of the DCSs using the IAM and screening corrected independent atom model (IAMS) for the scattering of electrons and positrons by $C_{2}H_{6}$ are displayed in Figure 1, Figure 2, Figure 3, Figure 4, Figure 5, Figure 6 and Figure 7. The electron impact DCSs are calculated in the energy range $E_{i}=$ 17.5 eV–1 MeV and positron impact DCSs for $E_{i}=$ 5 eV–1 MeV. As evident from Figure 1, Figure 2, Figure 3 and Figure 4, the IAMS method significantly reduces the DCSs at lower electron energies and lower scattering angles than those obtained by the IAM method. The lower DCSs, produced by the IAMS, are consistent with the experimental cross-sections. Thus, the inclusion of screening correction in the IAM has a substantial effect in the DCS, particularly at incident energies up to 100 eV. At higher energies, the two methods, as expected, produce almost identical DCSs (Figure 2, Figure 3 and Figure 4). At large energies, the de Broglie wavelength of the projectile is small enough so that the projectile interacts only with the constituent atoms and there is no overlapping effect on the DCSs. At lower electron energies, interference structures are observed in the angular distribution of electrons scattered elastically from C${}_{2}$H${}_{6}$. These structures, sensitive to the collision dynamics and the projectile–target interaction, are the interference effects caused by the leptons scattering from the constituent atoms. At high incident energies, the waning of these structures is observed. As a result, the DCS decreases monotonously with energy, without yielding any maximum or minimum. This waning is due to the incoherent interference of the large number of angular momentum states.

In Figure 1, Figure 2, Figure 3 and Figure 4, the DCSs, calculated using the IAM and IAMS, for electron scattering are compared with the experimental works of Curry et al. [20] for 17.5 and 20 eV; Tanaka et al. [22] for 20, 40, and 100 eV; Rawat et al. [23] for 40, 60, 80, 100, 200, 300, 400, and 500 eV; Fink et al. [16] for 100, 200, 400, 600, and 1000 eV; and Maji et al. [27] for 300, 500, 700, 1100, and 1300 eV projectile energy. The other theoretical calculations are compared with Rawat et al. [23] at 20, 40, 100, 200, 300, and 500 eV; Bettega et al. [33] at 20 eV; and Maji et al. [27] at 300, 500, 700, 900, 1100, and 1300 eV collision energy. There is no work found in the literature to compare with for the energies above 1300 eV, and the present is the only work at these energies for $e^{-}$–$C_{2}H_{6}$ scattering. The calculations at these energies were carried out to furnish future experiment for these energy points. As shown in Figure 1a,b, the calculated DCSs using IAM overestimate the cross-sections throughout the entire angular range, while those using IAMS fairly agree with other experimental measurements [20,22] and theoretical calculations [23,33] with slight differences in the pattern. Figure 1c,d show that the calculated DCSs using the IAMS agree both in pattern and magnitude with [22,23]. For energies $E_{i}\geq 100$ eV, both of our calculations produce almost identical results and agree well with the cross-sections available in the literature both in magnitude and pattern. The similar DCSs produced by both the IAM and IAMS at higher energies reflect the fact that the de Broglie wavelengths of the incident electron at these energies are small compared to the inter-atomic distances of the target for which the projectile interacts only with the constituent atoms, and there is no effect due to the overlapping of the orbitals. The low-energy failure of the IAM can be attributed to ignoring the redistribution of atomic electrons due to molecular binding and multiple scattering within the molecule and considering the constituent atoms as independent scatterers. However, the low energy failure of the IAMS is due to the semi-classical nature of the incorporated screening correction, the low energy limitation of the optical model, and the intrinsic low energy limitation of the IAM. From Figure 2e,f, we see that the experimental DCS of Rawat et al. [23] and Fink et al. [16] show disagreement with each other above ∼$60^{\circ }$. Moreover, from Figure 3, we see that the present theoretical model and that of Maji et al. [27] agree with the experimental DCS data of Fink et al. [16], but disagree with that of [27]. These disagreements depict the fact that the experimental data might contain ambiguities and theoretical models are required to overcome the ambiguities persisting in the experimental data, as well as helping to furnish future experiments.

There is no experimental measurement for the DCS found in the literature for the scattering of positrons by $C_{2}H_{6}$ to the best of our knowledge, and our calculated DCSs using the IAM and IAMS approaches were compared with the theoretical calculations of Chiari et al. [28] at 5 and 10 eV and the data generated by employing the additivity rule on the atomic DCS data of Dapor and Miotello [34] at 500, 1500, 2000, 2500, 3000, 3500, and 4000 eV collision energy. The comparison made in Figure 5a,b clearly depicts that our calculated DCSs and those by Chiari et al. [28] disagree significantly, both in magnitude and pattern. This disagreement might be due to the choice of different procedures and potentials. More data, both experimental and theoretical, are needed to bring out the more acceptable illustration of the positron impact DCS. It is evident from Figure 6c,f and Figure 7a–c that both the IAM and IAMS produce similar DCSs, both in magnitude and pattern, as those by the AR of Dapor and Miotello [34] with slight underestimation at 500, 1000, and 1500 eV. Here, again, the cross-sections are reduced at lower scattering angles using the IAMS over the IAM. As mentioned earlier, there is only one work (theoretical) found in the literature on $e^{+}$–$C_{2}H_{6}$ DCSs; the present study is, therefore, extremely useful to benchmark a theory. The work of Chiari et al. [28] was limited to 10 eV projectile energy only, while the present study covered the calculation of DCSs over a wide range of incident energy (1 eV–1 MeV), as presented in Figure 5, Figure 6 and Figure 7. Thus, the present work can be useful to furnish future experiments on positron scattering by $C_{2}H_{6}$.

The knowledge of spin polarization or the Sherman function S(θ) is important for projectile–molecule scattering, as it can furnish the details of the scattering process. As such, in the present study, we investigated the spin polarization for the scattering of electrons and positrons by $C_{2}H_{6}$ over the collision energy $E_{i}=10$–1500 eV in terms of the IAM approach. Figure 8 displays the spin polarization for the scattering of electrons, while that for positrons is shown in Figure 9. The predicted spin polarizations were compared with the calculations obtained by applying the additivity rule on the atomic spin polarization calculations of Fink and Yates [30]. In Figure 8a, a rapid change in sign and magnitude of S(θ) is observed near ∼100${}^{\circ }$ that matches the minimum observed in the electron impact DCS at 20 eV. This rapid change in the value of the Sherman function is observed near a minimum in the angular distribution of electrons scattered elastically [68]. Thus, the predicted Sherman function manifests the behavior of the experimental observation. We see from Figure 8 that after $100^{\circ }$, no rapid change of sign occurs in S(θ). There is no study found in the literature for positrons spin polarization, and the present work is the first one to calculate the S(θ) for $e^{+}$–$C_{2}H_{6}$ scattering. The sign uniformity over the whole angular range and the small magnitude of the positron impact Sherman function indicates that the positron impact DCS does not contain a significant extremum.

The energy dependence of the DCS and S(θ) for electron scattering at scattering angles $30^{\circ }$, $90^{\circ }$, and $150^{\circ }$ is shown in Figure 10. These angles were chosen to cover both forward and backward scattering. The comparison made in Figure 10a,b clearly shows the success of the IAMS over the IAM in producing the DCS for electron scattering. While the IAM produces an overestimated DCS, particularly at lower projectile energies up to about 100 eV, the IAMS results closely resemble the experimental measurements of [16,20,22,23,24,27] at $30^{\circ }$ and $90^{\circ }$ angle of scattering. As is evident from Figure 10d,e, the value of the electron S(θ) at $30^{\circ }$ becomes highest as 0.0001, corresponding to 30 eV at $90^{\circ }$ and S(θ) becomes maximum as 0.0015, corresponding to 8 eV incident energy. At $150^{\circ }$ angle of scattering, S(θ) increases slightly with energy up to 60 eV, then decreases monotonously up to $10^{6}$ eV projectile energy. The energy dependence of the positron DCS and S(θ) is shown in Figure 11 at the same scattering angles of $30^{\circ }$, $90^{\circ }$, and $150^{\circ }$ and compared to the theoretical calculations of Chiari et al. [28] and Dapor and Miotello [34]. Our calculated cross-sections using the IAM and IAMS do not agree in pattern and magnitude with the work of Chiari et al. [28]. On the other hand, both of our calculations agree well with [34] for the DCS of $e^{+}$–$C_{2}H_{6}$ scattering. At $150^{\circ }$, the S(θ) becomes steady up to 300 eV, then increases smoothly with the incident energy.

### 3.2. TCS, TICS, IECS, INCS, MTCS, and VCS

Figure 12 and Figure 13 present the calculations for the TCS, TICS, IECS, INCS, MTCS, and VCS for $e^{\pm }$–$C_{2}H_{6}$ scattering in the energy range $E_{i}=$1 eV–1 MeV using the IAM and IAMS in the framework of the OM potential. The calculated scattering observables for electron scattering were compared with the experimental measurements of Floeder et al. [19], Sueoka and Mori [21], Nishimura and Tawara [25], Ariyasinghe and Powers [29], Szmytkowski and Krzysztofowicz [26], Grill et al. [35], Duric et al. [36], Tian and Vidal [37], Schram et al. [38], Chatham et al. [69], Nishimura and Tawara [39], Rawat et al. [23], Tanaka et al. [22], and Shishikura et al. [40]; and the theoretical calculations of Joshipura and Vindkumar [42], Jin-Feng et al. [43], Vinodkumar et al. [44], Rawat et al. [23], Hwang et al. [45], Mayol and Salvat [70], Bettega et al. [33,41], Sun et al. [32], and Hayashi [46]. As is evident from Figure 12a,c–f, the calculated TCS, IECS, INCS, MTCS, and VCS using the IAM overestimate the experimental data, particularly at lower electron energies than those using the IAMS and other studies. The theoretical result of Rawat et al. [23] using the single-center-expansion close-coupling framework with Padé’s correction can correctly reproduce the cross-sections at lower projectile energy. The electron impact TCS seems to be a closed problem for the $e^{\pm }$–C${}_{2}$H${}_{6}$ system. One of the reasons behind this good prediction of the TCS is that several nonunique DCSs can lead to the same total elastic scattering cross-section after the integration over the angles [71].

The IAM and IAMS generated TCS, TICS, IECS, INCS, MTCS, and VCS for the scattering of positrons by $C_{2}H_{6}$ are compared in Figure 13 with the measurements of Chiari et al. [28], Sueoka and Mori [21], and Floeder et al. [19]; and theoretical studies of Singh and Antony [49], Raizada and Baluja [47], Raj and Tomar [48], Chiari et al. [28], Occhigrossi and Gianturco [50], and Dapor and Miotello [34]. Here, again, the IAM overestimates the experimental TCS, while the IAMS generated cross-sections fairly agree with the experimental data [19,21,28] and the theoretical calculations of [47,48,49] (Figure 13a). As shown in Figure 13c, the IAMS, Chiari et al. [28], Occhigrossi and Gianturco [50], and Dapor and Miotello [34] produce similar cross-sections, while the IAM result overestimates the IECS, especially at lower incident energies. Although the calculated MTCS using the IAM and IAMS and the TICS, using the IAMS, are consistent in pattern with the result of Singh and Antony [49], but different in magnitude. Figure 13 e,f show that both the IAM and IAMS results for the MTCS and VCS agree with the calculations of Dapor and Miotello [34]. In Figure 13d, we present our results for the positron INCS, which shows that both the IAM and IAMS produce similar cross-sections, both in pattern and magnitude.

### 3.3. Comparison of DCS and S(θ) for Electron and Positron Scattering

In Figure 14, the electron impact observable quantities are compared with those for positron impact. It is evident from Figure 14a that the DCSs for electron scattering were found to be higher in magnitude than those for positron scattering up to ∼2000 eV. Beyond this energy, the electron and positron DCSs have almost the same values. The CPP and the static potential (SP) hold the same sign during electron interaction and the opposite sign during positron interaction. Due to the opposite sign, these two potentials show annulment behavior of each other in the positron interaction. For this cancellation tendency of the CPP and SP and the absence of the exchange potential, the positron impact DCS is smaller than that of the electron impact at low energies. At high energies, the effect of polarization and exchange decreases and the static potential starts to dominate. As a result, the difference between the high-energy cross-sections for electrons and positrons diminishes. It is seen from Figure 14b that the electron impact S(θ) exhibits some interference structures and the positron impact S(θ), smooth behavior at lower incident energies. Perhaps this is due to the exchange potential. Anti-symmetry between the electron and positron impact Sherman function is observed at high incident energy. The higher value of S(θ) for positron scattering at higher energies compared to that for electron scattering is due to the effect of the Coulomb potential. This finding of large S(θ) at higher energies for positron scattering is in agreement with the work of Akter et al. [53] for ${\mathrm{NH}}_{3}$. The comparison of the (c) TCS, (d) TICS, (e) IECS, and (f) INCS between the scattering of electrons and positrons is also displayed in Figure 14. The effect of the aforementioned causes is clearly visible at lower incident energies, where the cross-sections for electron scattering were found to be different both in magnitude and pattern. It was also observed that most of the differences were observed within ∼1000 eV. At higher energies, the scattering observables follow a similar trend for both electron and positron scattering.

## 4. Conclusions

The Dirac partial wave theory (DPWT) in the context of optical model (OM) potential was used to calculate the elastic DCS, S(θ), TCS, TICS, IECS, INCS, MTCS, and VCS for the scattering of $e^{\pm }$ by the $C_{2}H_{6}$ molecule over the energy range $E_{i}=1\;\mathrm{eV}$–1 MeV. The derived scattering observables were compared to the available experimental measurements and theoretical calculations found in the literature. We used two approaches, the IAM and IAMS, to calculate the cross-sections for both electron and positron scattering. The calculated results for the DCS in the $e^{-}$–C${}_{2}$H${}_{6}$ scattering satisfactorily agree with the experimental data at collision energies $E_{i}\geq 60$ eV. The calculated results of the TCS, TICS, IECS, INCS, MTCS, and VCS using the IAMS also fairly agree with other experimental and theoretical works, except at very low energy. The low energy failure of the IAM can be attributed to ignoring the redistribution of atomic electrons due to molecular binding and multiple scattering within the molecule and considering the constituent atoms as independent scatterers. Sophisticated quantum mechanical methods such as the ab initio, *R*-matrix, or coupled channels methods can overcome this limitation, but these methods are not easy to implement and are also time consuming. For positron scattering, there is only one theoretical work and no experimental data are available to compare with. To the best of our knowledge, the DCSs for some energy points were calculated for the first time in the present work. The spin polarization describing the degree of polarization of the unpolarized projectiles was calculated for the first time in the present study for the $e^{\pm }$–$C_{2}H_{6}$ collision. Therefore, the present study on the scattering of positrons by the $C_{2}H_{6}$ molecule is very important to furnish future experiments on this collision.

The addition of the screening correction to the IAM significantly reduces the cross-sections at lower incident energies and at lower angles of scattering, thereby producing improved results. At higher energies, the target molecule becomes fully transparent to the incident lepton due to the smaller value of the de Broglie wavelength. Therefore, considering the constituent atoms as independent scatterer causes no significant error. Moreover, high energy atomic cross-sections are not large enough to become overlapped. Therefore, the effect of screening correction diminishes at high incident energies and IAM and IAMs provide almost the similar cross-sections, both in magnitude and pattern. The ability of the IAM method with screening correction shows its success in generating the cross-section data and spin polarization for electrons and positrons scattered by $C_{2}H_{6}$. Therefore, the IAMS using the DWPT with OM potential was proven to be successful in correctly describing the collision dynamics of $e^{\pm }$–$C_{2}H_{6}$ scattering over a wide range of projectile energies. From the available data, both experimental and theoretical, it seems that the electron impact TCS might be a closed problem for the $e^{\pm }$–$C_{2}H_{6}$ scattering system. One of the reasons behind this fact is that several nonunique DCSs can lead to the same total elastic scattering cross-section after the integration over the angles [71]. Moreover, the DCS is the more fundamental physical quantity to test the collision theory [71]. From the comparison of the data, both experimental and theoretical, we see that the experimental DCS data might contain ambiguities. Therefore, a sound theoretical model, particularly a single easy-to-use model capable of predicting a wide spectrum of observable quantities over a wide energy range, always carries greater importance in collision dynamics. In view of these facts, it can be concluded that our screening-corrected recipe might be useful to mitigate the partial demand of the observable quantities related to the scattering of $e^{\pm }$ off the ethane molecule in many research and technical fields. More data are needed for further refinement of the theory. 

## Figures and Tables

**Figure 1 molecules-28-01255-f001:**
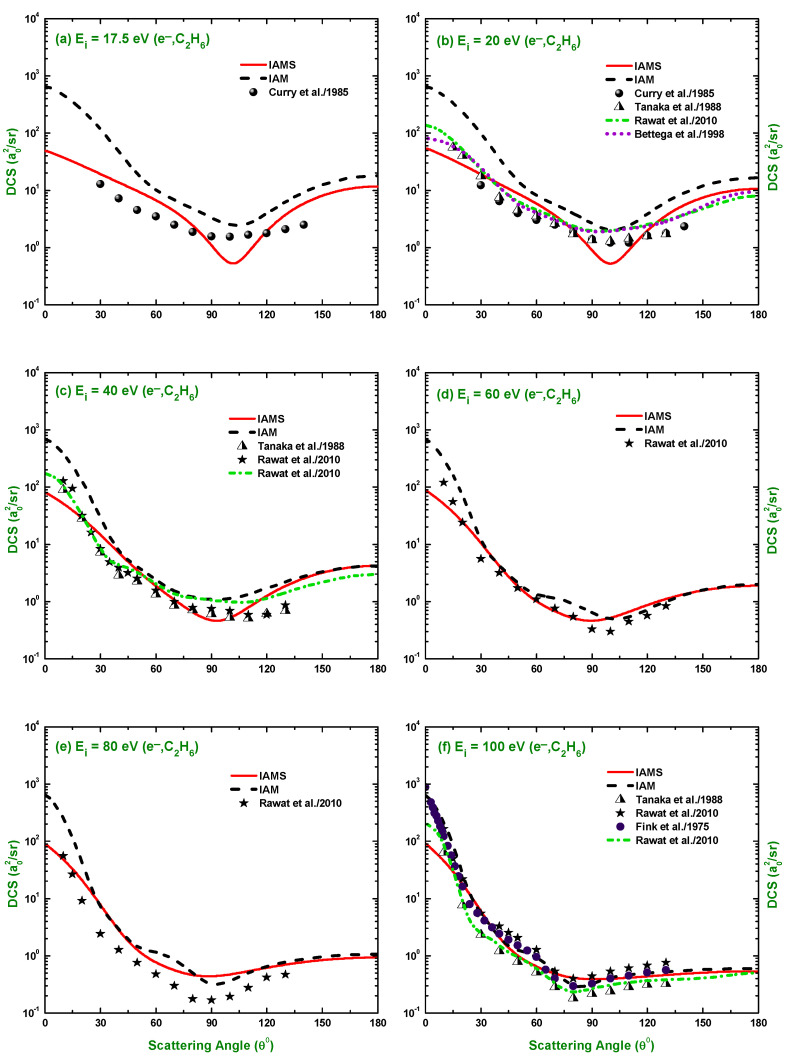
DCS $\left(a_{0}^{2}/sr\right)$ for the elastic scattering of electrons from $C_{2}H_{6}$ at energies of 17.5, 20, 40, 60, 80, and 100 eV. Theoretical works: IAM, IAMS, Rawat et al. [23] and Bettega et al. [41]; Experimental works: Curry et al. [20], Tanaka et al. [22], Rawat et al. [23], and Fink et al. [16].

**Figure 2 molecules-28-01255-f002:**
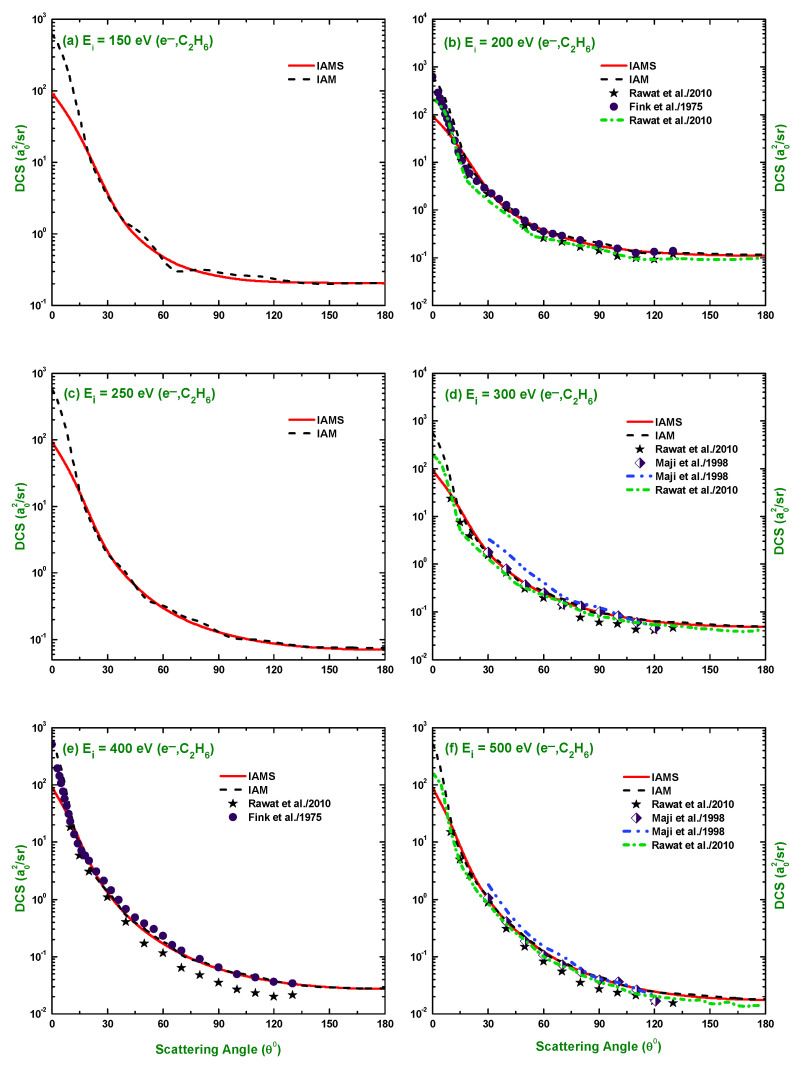
DCS $\left(a_{0}^{2}/sr\right)$ for the elastic scattering of electrons from $C_{2}H_{6}$ at energies of 150, 200, 250, 300, 400, and 500 eV. Theoretical works: References in Figure 1 and Maji et al. [27]; Experimental works: References in Figure 1 and Maji et al. [27].

**Figure 3 molecules-28-01255-f003:**
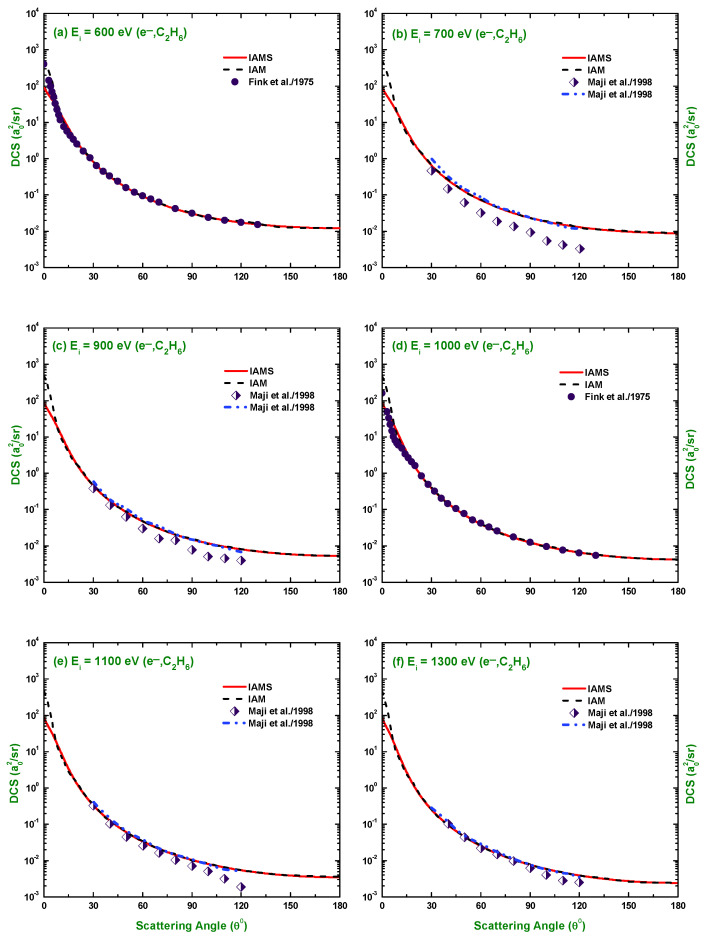
DCS $\left(a_{0}^{2}/sr\right)$ for the elastic scattering of electrons from $C_{2}H_{6}$ at energies of 600, 700, 900, 1000, 1100, and 1300 eV. Theoretical works: References in Figure 1 and Figure 2; Experimental works: References in Figure 1 and Figure 2.

**Figure 4 molecules-28-01255-f004:**
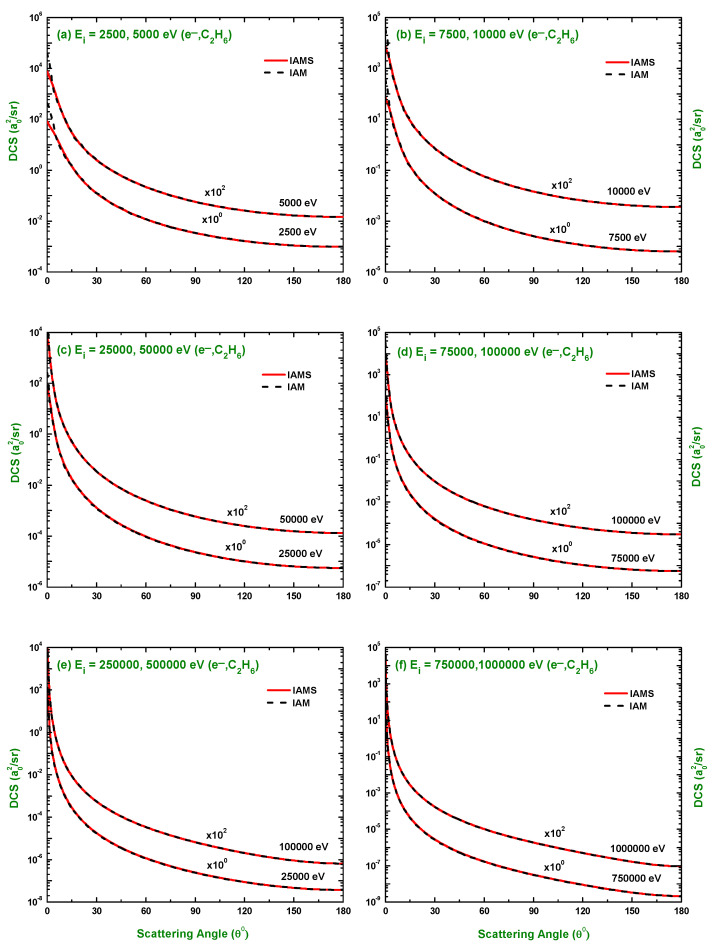
DCS $\left(a_{0}^{2}/sr\right)$ for the elastic scattering of electrons from $C_{2}H_{6}$ at energies of 2.5 keV to 1 MeV. Theoretical works: References in Figure 1.

**Figure 5 molecules-28-01255-f005:**
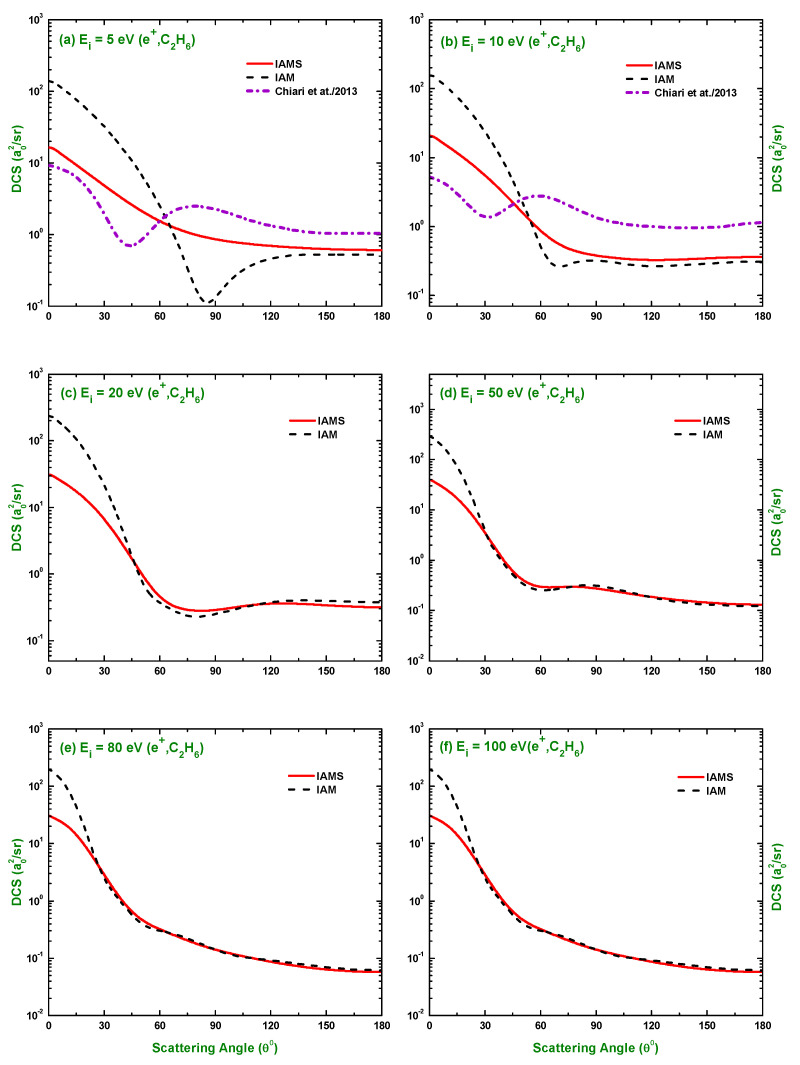
DCS $\left(a_{0}^{2}/sr\right)$ for the elastic scattering of positrons from $C_{2}H_{6}$ at energies of 5, 10, 20, 50, 80, and 100 eV. Theoretical works: IAM, IAMS, and Chiari et al. [28].

**Figure 6 molecules-28-01255-f006:**
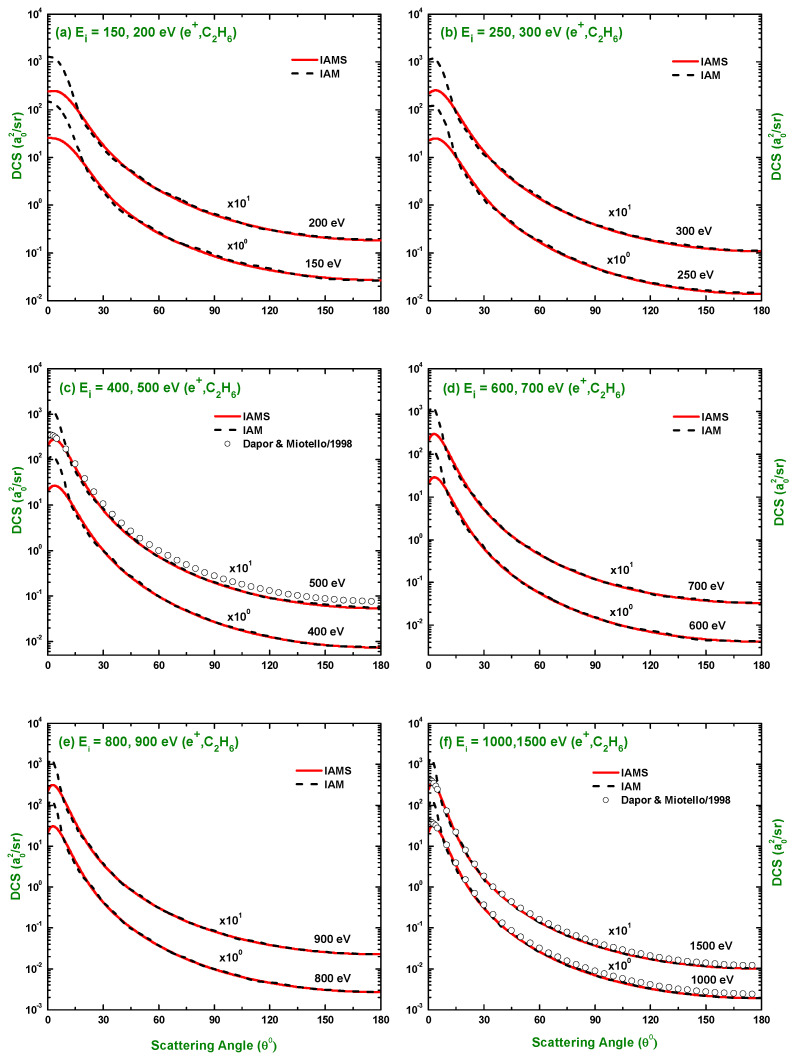
DCS $\left(a_{0}^{2}/sr\right)$ for the elastic scattering of positrons from $C_{2}H_{6}$ at energies of 150 eV to 1500 eV. Theoretical works: References in Figure 5 and Dapor and Miotello [34].

**Figure 7 molecules-28-01255-f007:**
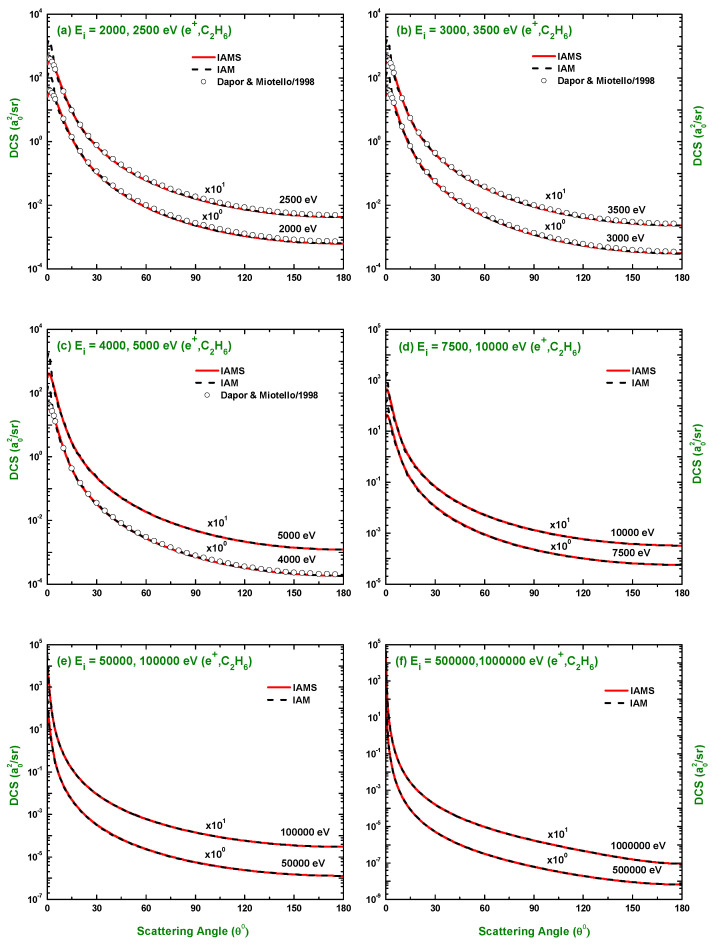
DCS $\left(a_{0}^{2}/sr\right)$ for the elastic scattering of positrons from $C_{2}H_{6}$ at energies of 2.0 keV to 1 MeV. Theoretical works: References in Figure 6.

**Figure 8 molecules-28-01255-f008:**
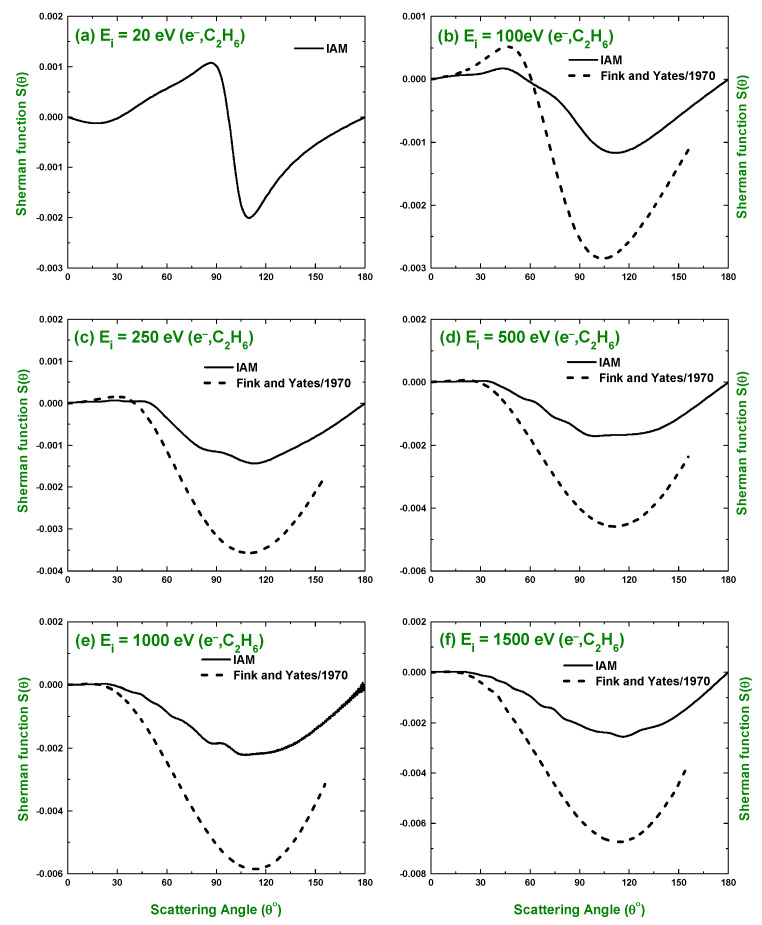
Angular distribution of the Sherman function for the electrons scattered elastically from C${}_{2}$H${}_{6}$ predicted by the IAM approach. Theoretical works: Fink and Yates [30].

**Figure 9 molecules-28-01255-f009:**
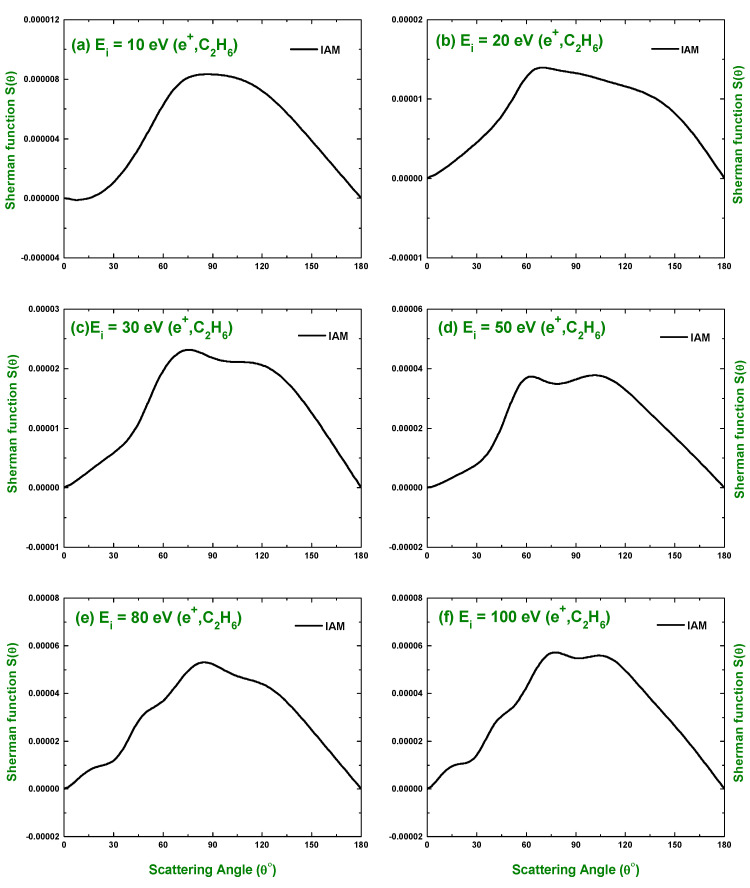
Angular distribution of the Sherman function for the positrons scattered elastically from C${}_{2}$H${}_{6}$ predicted by the IAM approach.

**Figure 10 molecules-28-01255-f010:**
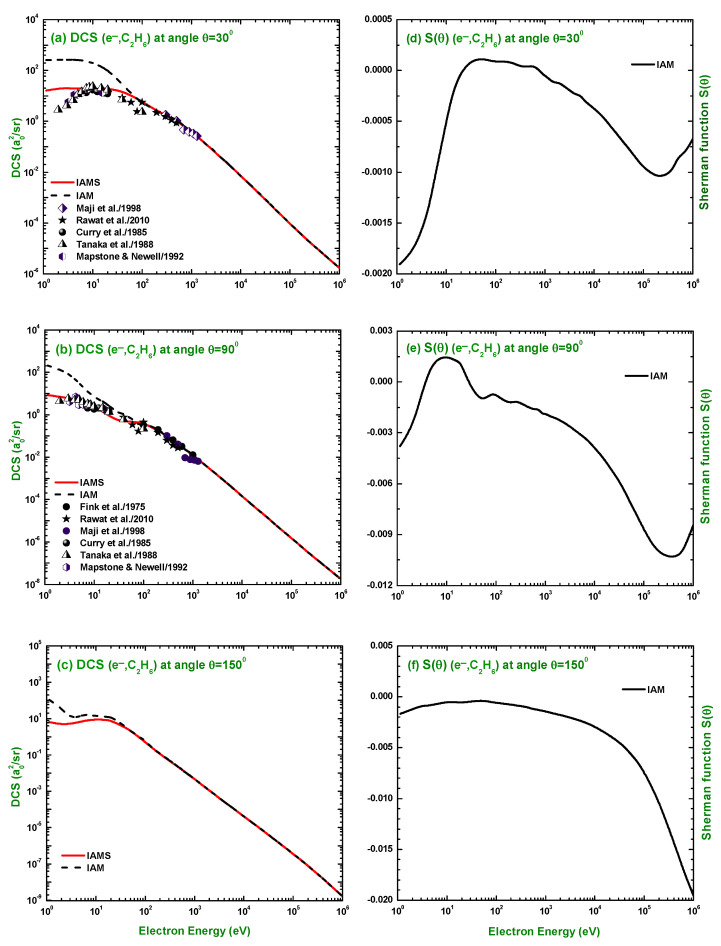
Energy dependence of DCS $\left(a_{0}^{2}/sr\right)$ and the Sherman function for the elastic scattering of electrons from $C_{2}H_{6}$ at angles $30^{\circ }$, $90^{\circ }$, and $150^{\circ }$. Theoretical works: IAM and IAMS; Experimental works: Maji et al. [27], Rawat et al. [23], Curry et al. [20], Mapstone and Newell [24], Tanaka et al. [22], and Fink et al. [16].

**Figure 11 molecules-28-01255-f011:**
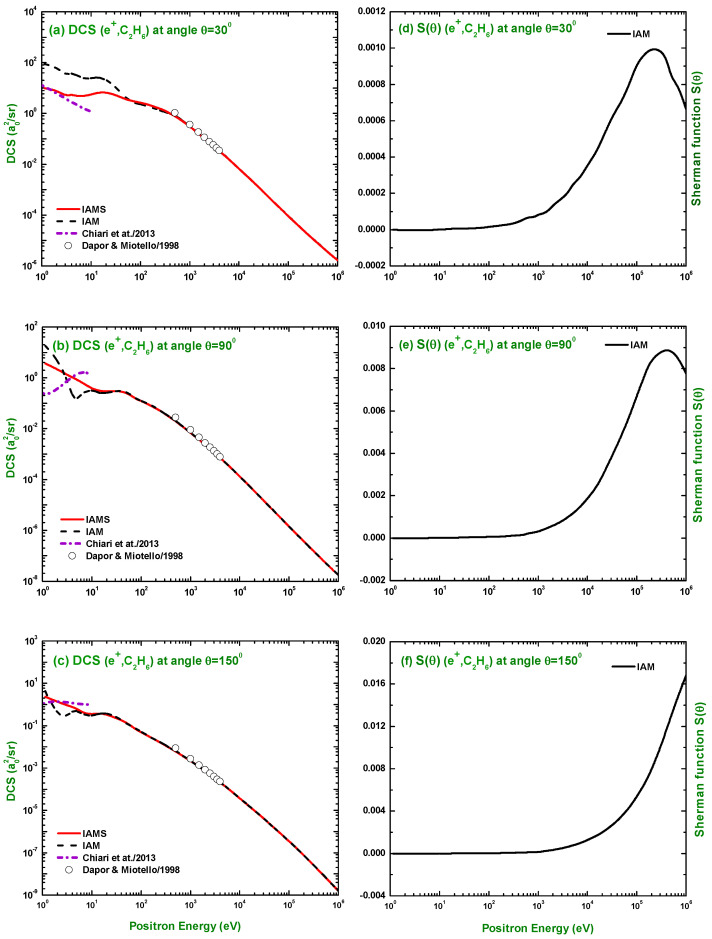
Energy dependence of DCS $\left(a_{0}^{2}/sr\right)$ and the Sherman function for the elastic scattering of positrons from $C_{2}H_{6}$ at angles 30${}^{\circ }$, 90${}^{\circ }$, and 150${}^{\circ }$. Theoretical works: IAM, IAMS, Chiari et al. [28], and Dapor and Miotello [34].

**Figure 12 molecules-28-01255-f012:**
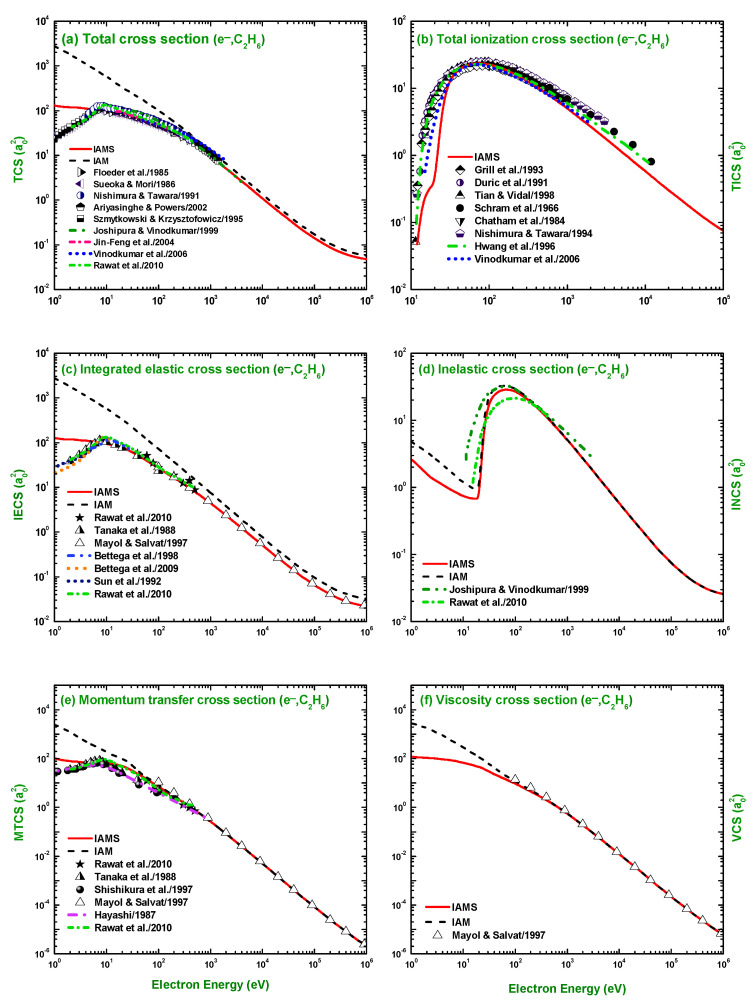
(**a**) TCS, (**b**) TICS, (**c**) IECS, (**d**) INCS, (**e**) MTCS, and (**f**) VCS $\left(a_{0}^{2}\right)$ for the scattering of electrons from $C_{2}H_{6}$. Theoretical works: IAM, IAMS, Joshipura and Vindkumar [42], Jin-Feng et al. [43], Vinodkumar et al. [44], Rawat et al. [23], Hwang et al. [45], Mayol and Salvat [70], Bettega et al. [33,41], Sun et al. [32], and Hayashi [46]; Experimental works: Floeder et al. [19], Sueoka and Mori [21], Nishimura and Tawara [25,39], Ariyasinghe and Powers [29], Szmytkowski and Krzysztofowicz [26], Grill et al. [35], Duric et al. [36], Tian and Vidal [37], Schram et al. [38], Chatham et al. [69], Rawat et al. [23], Tanaka et al. [22], and Shishikura et al. [40].

**Figure 13 molecules-28-01255-f013:**
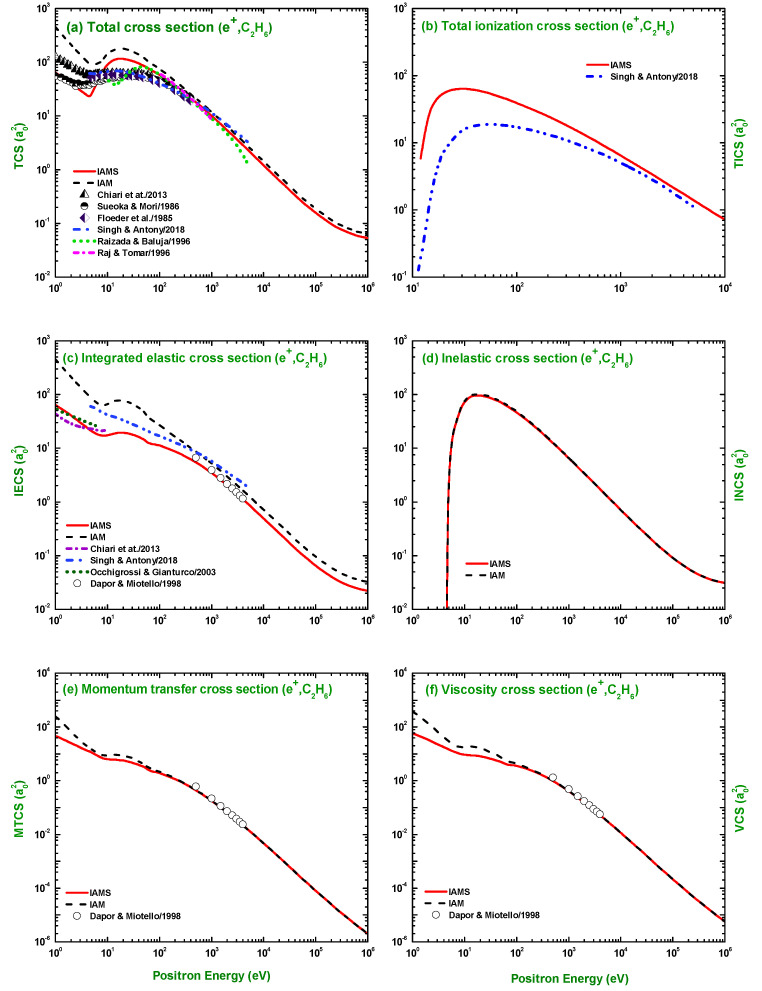
(**a**) TCS, (**b**) TICS, (**c**) IECS, (**d**) INCS, (**e**) MTCS, and (**f**) VCS $\left(a_{0}^{2}\right)$ for the scattering of positrons from $C_{2}H_{6}$. Theoretical works: IAM, IAMS, Singh and Antony [49], Raizada and Baluja [47], Raj and Tomar [48], Chiari et al. [28], Occhigrossi and Gianturco [50], and Dapor and Miotello [34]; Experimental works: Chiari et al. [28], Sueoka and Mori [21], and Floeder et al. [19].

**Figure 14 molecules-28-01255-f014:**
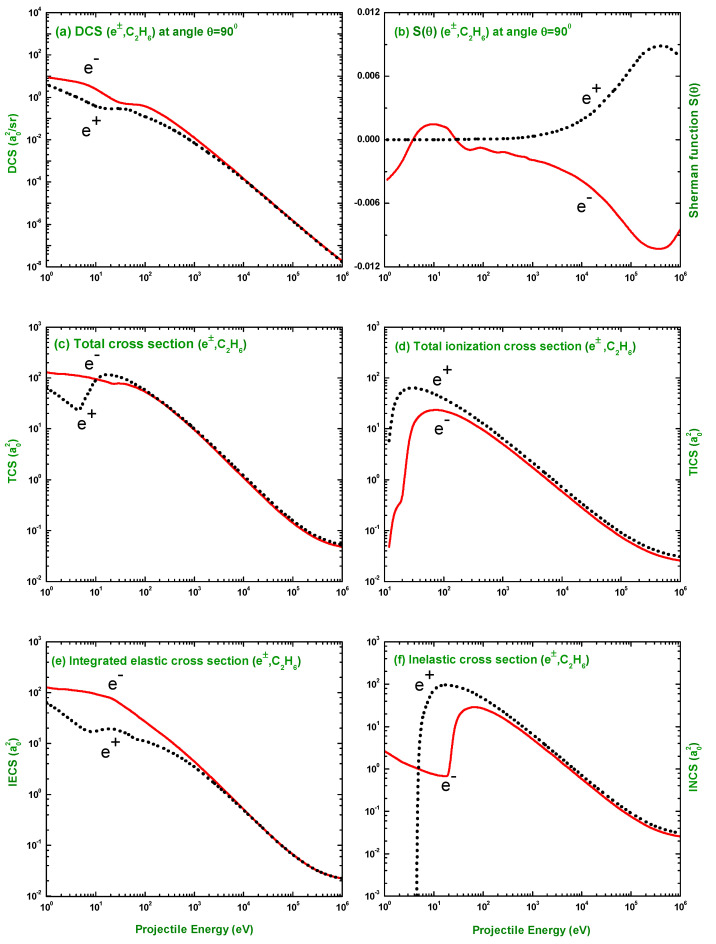
Comparison of (**a**) DCS $\left(a_{0}^{2}/sr\right)$, (**b**) Sherman function, (**c**) TCS, (**d**) TICS, (**e**) IECS, and (**f**) INCS $\left(a_{0}^{2}\right)$ for the scattering of electrons and positrons from $C_{2}H_{6}$. Theoretical: Sherman Function (IAM) and DCS, TCS, TICS, IECS and INCS (IAMS).

**Table 1 molecules-28-01255-t001:** Target properties of C${}_{2}$H${}_{6}$ [49] ($\alpha_{\mathrm{mol}}$—molecular polarizability, IP—ionization potential, and $\Delta_{e}$—first electronic excitation energy).

$\alpha_{\mathrm{mol}}$ (Å${}^{3}$)	IP (eV)	$\Delta_{e}\left(\mathrm{eV}\right)$
4.226	11.52	10

**Table 2 molecules-28-01255-t002:** Coordinate geometry of C${}_{2}$H${}_{6}$ in Cartesian coordinate system (Angstroms) [67].

Atom	x	y	z
C	0.0000	0.0000	0.7622
C	0.0000	0.0000	−0.7622
H	0.0000	1.0189	1.1572
H	−0.8824	−0.5094	1.1572
H	0.8824	−0.5094	1.1572
H	0.0000	−1.0189	−1.1572
H	−0.8824	0.5094	−1.1572
H	0.8824	0.5094	−1.1572

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
