# Peer review of "Scattering of e± by C2H6 Molecule over a Wide Range of Energy: A Theoretical Investigation"

_molecules, 2023, doi:10.3390/molecules28031255_

Round 1

Reviewer 1 Report

The manuscript “Scattering of e+- by C2H6 molecule over a wide range of energy: a theoretical investigation” by Sathee et al. reports results of theoretical calculations for electron and positron scattering by ethane. This is a somewhat simple but very important target, which was relatively well studied for electron scattering but on the positron case, the studies are somewhat scarce. Thus, for sure a work aiming to fulfill this gap in positron scattering studies is worth of publication.  

In the introduction the authors present some applications for positron physics. But most of these applications are based in the positron annihilation, which is not the scope of this work. Thus, the authors should expand this paragraph of introduction discussing the applicability of the data reported by them.  

In the 7th paragraph of Introduction, the authors justify the electron-scattering study in the absence of results for Sherman functions for this target. It would be very helpful to include a brief explanation for these functions here, explaining, for example, which information or conclusions can be gotten from the Sherman functions that cannot be accessed from any other result. 

In the next paragraph the authors discuss about the uncertainties in experiments and the importance of a “sound theory” in order to derive consistent cross sections. This is true, but the authors should also include some discussion about the “uncertainties” in the calculations (are all calculations converged for different basis sets? model potentials? Molecular geometry?). 

The authors should define the ELSEPA acronym. 

Which is the reason to present the calculated DCS with different energy ranges for electron and positron scattering? (from 5 eV for positron and 17.5 eV for electron). 

Previous calculations should be present as lines in the figures 7, 8, 10 and 11  

In the results the authors argue that the Sherman functions can provide details in the scattering process, but their results are merely presented and not properly discussed. The authors should expand this discussion or remove these figures from the text. 

In figure 15 there are some sets of previous calculations that are hidden by experimental data. Using empty symbols or putting the lines in front of the symbols could resolve this issue. 

In summary, this is a relevant work, in particular for fulfilling the gap in positron-molecule scattering. But it only could be published after the authors deal with the issues listed above.

Reviewer 2 Report

The manuscript “Scattering of e+/- by C2H6 molecule over a wide range of energy: A theoretical investigation” by Sathee et al. reported theoretical studies of electron and positron scattering with ethane molecule over a wide range of energy (1 eV – 1 MeV). Partial-wave decomposition of the scattering states within the Dirac relativistic framework employing a free-tom complex optical model potential is used to calculate the scattering processes, which obtain various types of cross sections, i.e. differential, integrated elastic, momentum transfer, viscosity, inelastic, grand total and total ionization cross sections. The results are compared with the experimental and theoretical works available in the literature. The addition of the screening correction to the independent atom approximation method has been found to well describe the collision dynamics of e+/- C2H6 scattering over a wide range of projectile energies. These results are important to provide the basic understanding of electron/positron transport phenomena and interaction processes with atoms and nuclei of the target as well as important insights into the collision dynamics and molecular structure. I recommend to publish this work in Molecules with some minor comments:

 A brief description about the differences between electron and positron scattering cross sections is needed in the Results and Discussion part.

 It is suggested to present the data also in a tabular form (possibly in Supplementary material), which can be helpful for the readers.

 Line 5: “obervable” should be “observable”

 Line 55: “play important role” should be “play an important role” or “play important roles”

 The authors are suggested to check more carefully on the English language (grammar, logic ...).

Round 2
